# ADA+: A GENERIC FRAMEWORK WITH MORE ADAPTIVE EXPLICIT ADJUSTMENT FOR LEARNING RATE

## ABSTRACT

Although adaptive algorithms have achieved significant success in training deep neural networks with faster training speed, they tend to have poor generalization performance compared to SGD WITH MOMENTUM(SGDM). One of the state-of-the-art algorithms, PADAM, is proposed to close the generalization gap of adaptive methods while lacking an internal explanation. This work proposes a general framework, in which we use an explicit function $\Phi(\cdot)$ as an adjustment to the actual step size, and present a more adaptive specific form ADAPLUS(ADA+). Based on this framework, we analyze various behaviors brought by different types of $\Phi(\cdot)$, such as a constant function in SGDM, a linear function in ADAM, a concave function in PADAM and a concave function with offset term in ADAPLUS. Empirically, we conduct experiments on classic benchmarks both in CNN and RNN architectures and achieve better performance(even than SGDM). The code is anonymously provided in `https://anonfiles.com/daV7Ed6enb/AdaPlus_zip`.

## 1 INTRODUCTION

First-order optimization algorithms play a significant role in training deep neural networks. One of the dominant methods is SGD(Robbins & Monro, 1951), which updates parameters in a very concise form. SGD WITH MOMENTUM(Polyak, 1964) is a momentum-based improvement of SGD that combines raw gradients to reduce the interference of noise gradients and has proved itself to be an efficient first-order optimization algorithm. Besides, adaptive variants of SGD, such as ADAGRAD (Duchi et al., 2011), RMSPROP(Tieleman & Hinton, 2012), ADAM(Kingma & Ba, 2015), have recently emerged and achieved success due to their convenient and fast automatic learning rate adjustment mechanisms. However, since ADAM is indicated not to converge in certain instances, AMSGRAD(Reddi et al., 2018) suggests maintaining a non-increasing step size by a maximum historical term, so as to deal with the non-convergence problem.

On the one hand, adaptive methods are widely applied due to their quicker convergence at an early stage. However, there remain potential perils of adaptivity (Wilson et al., 2017), which suggest worse performance than the fine-tuned SGD/SGDM. Therefore, many architectures in the field of computer vision, such as VGGNet(Simonyan & Zisserman, 2015), ResNet(He et al., 2016), DenseNet(Huang et al., 2017), and natural language processing tasks(Klein et al., 2017), would be inclined to choose SGD-like algorithms to act as an optimizer. On the other hand, the 'small learning rate dilemma'(Chen & Gu, 2018) is also a challenge for adaptive methods, that is, it is difficult to apply the well-worked learning rate decaying strategy in SGD to adaptive gradient methods. With better performance, the PADAM algorithm(Chen & Gu, 2018) is proposed to solve this problem. However, the motivation of this *partially* adaptive momentum fails to explain the intrinsic implication of doing grid search for the power $p$ of the second-order moment estimation, along with the relationship between the desired learning rate and the *partially* adaptive term.

In this work, we first propose an adaptive adjustment function, which is a map based on historical gradient information to adjust the actual step size, and a framework that can explain the above phenomenon explicitly. Through our analysis of this function, which combines the SGD-like algorithm with the ADAM-like algorithm, we further explain the different behaviors of different algorithms. Moreover, we propose a brief but efficient specific form, called ADAPLUS(ADA+), with better adaptability to cope with the aforementioned challenges effectively. Empirically, we conduct

experiments on popular models and tasks both in CNN and RNN architectures with ADAPLUS. Comparing favorably to SGD-like approaches, it gains faster convergence, reduced oscillation and better performance. Compared with ADAM-like methods, it can adapt to a variable learning rate schedule, thus achieving significant improvement in performance in complex deep neural network architectures.

## 2 PRELIMINARIES AND MOTIVATIONS

### 2.1 NOTATIONS

As is generally agreed, we use lowercase letters $a$ for scalars, lowercase bold letters $\boldsymbol{a}$ for vectors, and uppercase bold letters $\boldsymbol{A}$ for matrices. Then, we set $\boldsymbol{\theta}_t \in \mathbb{R}^d$ as a parameter vector of a sequence in $d$-dimensional space, and use the scalar $\theta_{t,i}$ to represent the $i$-th elements in vector $\boldsymbol{\theta}_t$. The set of all positive definite $d \times d$ matrices is denoted by $\mathcal{S}_+^d$. Besides, operators are defined as following. For vectors $\boldsymbol{a}$ and $\boldsymbol{b}$ in the same dimension, we use $\sqrt{\boldsymbol{a}}$ to donate element-wise squares, $\boldsymbol{a}/\boldsymbol{b}$ to donate element-wise divisions. These notations are the same for matrices. For a given matrix $\boldsymbol{A} \in \mathcal{S}_+^d$ and a vector $\boldsymbol{b} \in \mathbb{R}^d$, we use $\Pi_{\mathcal{X},\boldsymbol{A}}(\mathbf{b})$ as a projection operator which means $\arg\min_{\boldsymbol{a} \in \mathcal{X}} \left\| \boldsymbol{A}^{1/2}(\boldsymbol{a} - \boldsymbol{b}) \right\|$, where $\mathcal{X}$ is a convex set which has a bounded diameter $D_\infty$, i.e., $\|\boldsymbol{a} - \boldsymbol{b}\|_\infty \leq D_\infty, \forall\, \boldsymbol{a}, \boldsymbol{b} \in \mathcal{X}$. Additionally, with a mild abuse of the notation of norms, we represent the $L^p$ norm formation of an exponential moving average (EMA) of $\boldsymbol{g}_t$ like this:

$$
\begin{aligned}
\|\boldsymbol{g}_t\|_p &\triangleq \boldsymbol{v}_t^{\frac{1}{p}} \\
&= [\beta_2^p \boldsymbol{v}_{t-1} + (1 - \beta_2^p)\, |\boldsymbol{g}_t|^p]^{\frac{1}{p}} \\
&= \left[ (1 - \beta_2^p) \sum_{i=1}^{t} \beta_2^{p(t-i)} \cdot |\boldsymbol{g}_i|^p \right]^{\frac{1}{p}}
\end{aligned}
\tag{2.1}
$$

### 2.2 MOTIVATIONS

As is described in Figure 1, for optimizers, we treat their adjustment term of $\alpha_t$ as a mapping from $\|\boldsymbol{g}_t\|_p$ to the actual adjustment value, i.e.,

$$
\boldsymbol{\theta}_{t+1} = \boldsymbol{\theta}_t - \frac{\alpha_t}{\Phi(\|\boldsymbol{g}_t\|_p)} \cdot \boldsymbol{m}_t
\tag{2.2}
$$

To avoid confusion, we call $\alpha_t$ the learning rate, which is a parameter that needs tuning and applying a decaying schedule. Apart from that, $\frac{\alpha_t}{\Phi(\|\boldsymbol{g}_t\|_p)}$ is called the actual step size. In this section, we summarize popular first-order stochastic optimization algorithms into the following four categories:

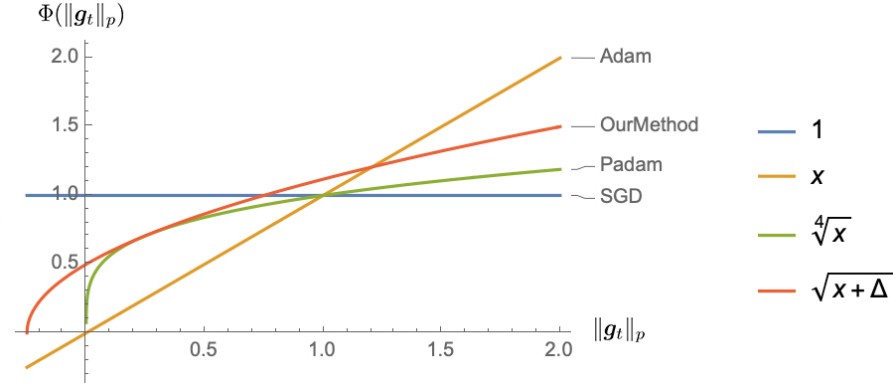

Figure 1: Different choices for $\Phi(\|\boldsymbol{g}_t\|_p)$, changing from Adam to SGD. We propose a specific function $\Phi(\|\boldsymbol{g}_t\|_p) \triangleq \sqrt{\|\boldsymbol{g}_t\|_p + \Delta}$, which is shown above.

- Type I  **SGDM (Constant function)**:

$$\boldsymbol{\theta}_{t+1} = \boldsymbol{\theta}_t - \frac{\alpha_t}{1} \cdot \boldsymbol{m}_t, \ i.e., \ \Phi(\|\boldsymbol{g}_t\|_p) = 1. \tag{2.3}$$

It can be seen from the formula in SGD WITH MOMENTUM(SGDM)that the actual step size of each update equals to $\alpha_t$, which is applied without adaptive adjustments. The high learning rate in an early stage makes the loss function fluctuate significantly on the surface, and its learning speed is not as fast as adaptive optimization algorithms and their variants.

- Type II  **ADAM (Linear function)**:

$$\boldsymbol{\theta}_{t+1} = \boldsymbol{\theta}_t - \frac{\alpha_t}{\|\boldsymbol{g}_t\|_2} \cdot \boldsymbol{m}_t, \ i.e., \ \Phi(\|\boldsymbol{g}_t\|_2) = \|\boldsymbol{g}_t\|_2. \tag{2.4}$$

Extremes of the learning rate(Wilson et al., 2017) in ADAM-like algorithms have brought many issues. As is shown in Figure 1, when the gradient is very large, the function value grows greatly in a linear rate, which leads to a rapid decrease in the step size; on the other hand, once the gradient is extremely small (ie, when the norm of EMA of $\boldsymbol{g}_t$ approaches to 0), $\Phi(\|\boldsymbol{g}_t\|_p) = \|\boldsymbol{g}_t\|_p$ is very close to 0, which makes it more than essential to select a much smaller learning rate(a few orders of magnitude smaller than SGDM) to get a reasonable actual step size. However, the much smaller initial learning rate makes ADAM-like algorithms not flexible enough to adapt to the various learning rate schedule. Although their early convergence is faster, the final performance tends to be poor due to the small actual step size in the final stage. A proposed solution, ADAMW(Loshchilov & Hutter, 2019), trying to change the way the weight attenuation is updated, achieves a better generalization performance, but there still remains a gap between adaptive methods and SGD-like methods.

- Type III  **PADAM (Concave function)**:

$$\boldsymbol{\theta}_{t+1} = \boldsymbol{\theta}_t - \frac{\alpha_t}{\|\boldsymbol{g}_t\|_2^p} \cdot \boldsymbol{m}_t, \ i.e., \ \Phi(\|\boldsymbol{g}_t\|_2) = \sqrt[4]{\|\boldsymbol{g}_t\|_2}. \tag{2.5}$$

Note that the $p$ here means *partial* rather than $L^p$ norm, its default setting is $\frac{1}{4}$ [1].

PADAM has achieved superior performance(Chen & Gu, 2018) in computer vision experiments, but this algorithm only introduces a new hyper-parameter $p$ and does grid searches for it without realizing the essential reason for its improvement relative to ADAM. The internal cause is that a concave function is applied rather than the linear function in ADAM. Once $\varepsilon$ is extremely small and $\|\boldsymbol{g}_t\|_2 \in (0, \varepsilon)$, the mapping value of $\Phi(\cdot)$ would be much larger in PADAM than in ADAM; therefore, PADAM can adapt to larger learning rate $\alpha_t$, thus flexibly adapting to the variable learning rate scheme.

- Type IV  **ADAPLUS(ADA+) (Concave function with offset)**:

$$\boldsymbol{\theta}_{t+1} = \boldsymbol{\theta}_t - \frac{\alpha_t}{\sqrt{\|\boldsymbol{g}_t\|_1 + \Delta}} \cdot \boldsymbol{m}_t, \ i.e., \ \Phi(\|\boldsymbol{g}_t\|_1) = \sqrt{\|\boldsymbol{g}_t\|_1 + \Delta}. \tag{2.6}$$

In ADAPLUS, we extend the concave function by introducing an offset $\Delta$ and present a default setting as above. This form of $\Phi(\cdot)$ not only directly inherits advantages of PADAM, as is depicted in Figure 1, but also makes a better guarantee for larger learning rates. The offset $\Delta$ makes sure that $\Phi(\cdot)$ can altogether avoid the extreme situation. Even when $\|\boldsymbol{g}_t\|_1 \to 0$, a more extensive learning rate $\alpha_t$ is allowed. Besides, when $\|\boldsymbol{g}_t\|_1$ is relatively large, we can also adaptively constrain the updates and achieve 'naturally annealing' like ADAM, but more moderately. Thus, we reckon that the proposed algorithm has better adaptive performance than both ADAM-like and SGD-like algorithms, which is why we call it ADAPLUS(ADA+).

So far, we may surprisingly notice that this offset $\Delta$ is the $\varepsilon$ that we used to apply in optimizers indeed. Nevertheless, the previous $\varepsilon$ is only used to avoid dividing by zero to keep the numerical stability. In fact, through our analysis above, the design of offset $\Delta$ can effectively avoid the occurrence of extreme step sizes, thus saliently improving adaptivity and performance.

---

[1]It is a bit different from the formation in (Chen & Gu, 2018), but actually the same.

We present a default setting as above, due to its excellent performance and concise form; however, there would be more elegant forms remaining for us to explore in the future. For instance, we can replace the square function with $log(\cdot)$ or $tanh(\cdot)$ and so on, or tune it into a $p$-th power function to apply a PADAM-like grid search.

# 3 THE PROPOSED FRAMEWORK AND METHOD

## 3.1 GENERIC FRAMEWORK

As is shown in Section 2.2, we propose a generic framework that includes SGD-like and ADAM-like algorithms. With the adaptive adjustment function $\Phi(\|\boldsymbol{g}_t\|_p)$ to combine algorithms in an organized way, behavioral characteristics of different algorithms can be explicitly observed.

---

**Algorithm 1** GENERIC FRAMEWORK

---

**Input:** $\boldsymbol{\theta}_1 \in \mathcal{X}$; learning rate $\{\alpha_t\}_{t=1}^T$; momentum parameters $\{\beta_{1t}\}_{t=1}^T$, $\beta_2$; $L^p$ norm parameter $p$; function for step estimation $\Phi(\cdot)$.

1: Set $\boldsymbol{m}_0 = \boldsymbol{0}$, $\boldsymbol{v}_0 = \boldsymbol{0}$, $\widehat{\boldsymbol{v}}_0 = \boldsymbol{0}$
2: **for** $t = 1$ **to** $T$ **do**
3:     $\boldsymbol{g}_t = \nabla f_t(\boldsymbol{x}_t)$
4:     $\boldsymbol{m}_t = \beta_{1t}\boldsymbol{m}_{t-1} + (1 - \beta_{1t})\boldsymbol{g}_t$
5:     $\boldsymbol{v}_t = \beta_2\boldsymbol{v}_{t-1} + (1 - \beta_2)|\boldsymbol{g}_t|^p$
6:     $\widehat{\boldsymbol{v}}_t = \max(\widehat{\boldsymbol{v}}_{t-1}, \boldsymbol{v}_t)$
7:     // Depending on whether AMSGrad is desired, the previous line can be optionally annotated.
8:     $\boldsymbol{\theta}_{t+1} = \Pi_{\mathcal{X}, \mathrm{diag}\left(\Phi^{-1}(\widehat{\boldsymbol{v}}_t^{1/p})\right)} \left(\boldsymbol{\theta}_t - \frac{\alpha_t}{\Phi(\widehat{\boldsymbol{v}}_t^{1/p})} \cdot \boldsymbol{m}_t\right)$
9: **end for**
10: // We omit the bias-correction terms and other misc for clarity.

---

According to our analysis, different settings of the adaptive adjustment function $\Phi(\|\boldsymbol{g}_t\|_p)$ will lead to different behaviors of the algorithm, such as $\Phi(\|\boldsymbol{g}_t\|_p) = 1$ for SGDM and $\Phi(\|\boldsymbol{g}_t\|_p) = \|\boldsymbol{g}_t\|_p$ for ADAM. In practice, we can selectively decide whether or not to open AMSGRAD according to the actual situation. We will present a specific algorithm that has a concise form and demonstrates superior performance in Section 5.

## 3.2 SPECIFIC FORMATION FOR ADAPLUS

---

**Algorithm 2** ADAPLUS

---

**Input:** $\boldsymbol{\theta}_1 \in \mathcal{X}$; learning rate $\{\alpha_t\}_{t=1}^T$; momentum parameters $\{\beta_{1t}\}_{t=1}^T$, $\beta_2$; offset term $\boldsymbol{\Delta}$.

1: Set $\boldsymbol{m}_0 = \boldsymbol{0}$, $\boldsymbol{v}_0 = \boldsymbol{0}$, $\widehat{\boldsymbol{v}}_0 = \boldsymbol{0}$
2: **for** $t = 1$ **to** $T$ **do**
3:     $\boldsymbol{g}_t = \nabla f_t(\boldsymbol{x}_t)$
4:     $\boldsymbol{m}_t = \beta_{1t}\boldsymbol{m}_{t-1} + (1 - \beta_{1t})\boldsymbol{g}_t$
5:     $\boldsymbol{v}_t = \beta_2\boldsymbol{v}_{t-1} + (1 - \beta_2)|\boldsymbol{g}_t|$
6:     $\widehat{\boldsymbol{v}}_t = \max(\widehat{\boldsymbol{v}}_{t-1}, \boldsymbol{v}_t)$
7:     // Depending on whether AMSGrad is desired, the previous line can be optionally annotated.
8:     $\boldsymbol{\theta}_{t+1} = \Pi_{\mathcal{X}, \mathrm{diag}\left(\sqrt{\widehat{\boldsymbol{v}}_t + \boldsymbol{\Delta}}\right)} \left(\boldsymbol{\theta}_t - \frac{\alpha_t}{\sqrt{\widehat{\boldsymbol{v}}_t + \boldsymbol{\Delta}}} \cdot \boldsymbol{m}_t\right)$
9: **end for**
10: // We omit the bias-correction terms and other misc for clarity.

---

Here, we express the specific algorithm corresponding to the **Type IV** function mentioned in Section 2.2 as following. It is a special case of Algorithm 1.

$$\boldsymbol{\theta}_{t+1} = \boldsymbol{\theta}_t - \frac{\alpha_t}{\sqrt{\|\boldsymbol{g}_t\|_1 + \Delta}} \cdot \boldsymbol{m}_t, \ i.e., \ \Phi(\|\boldsymbol{g}_t\|_1) = \sqrt{\|\boldsymbol{g}_t\|_1 + \Delta}. \tag{3.1}$$

**Two main contributions** are made, including the concave function $\sqrt{x}$ and an offset term $\Delta$, so the algorithm can avoid extreme actual step sizes and achieve superior empirical results.

# 4 CONVERGENCE ANALYSIS OF ADAPLUS

In this section, we provide a convergence analysis based on a standard online convex optimization framework(Zinkevich, 2003). For each time step $t \in [T]$, there are sequences of parameters $\{\boldsymbol{\theta}_t\}_{t=1}^T$ and convex loss functions $\{f_t\}_{t=1}^T$. Let $\mathcal{X}$ be a bounded convex feasible set, which includes $\{\boldsymbol{\theta}_t\}_{t=1}^T$ and the optimal solution $\boldsymbol{\theta}^*$. The optimal solution $\boldsymbol{\theta}^*$ is defined to achieve empirical risk minimization(ERM), i.e., $\boldsymbol{\theta}^* = \underset{\boldsymbol{\theta} \in \mathcal{X}}{\operatorname{argmin}} \sum_{t=1}^T f_t(\boldsymbol{\theta})$. We use *regret* $R_T$ to donate the entire difference between $\sum_{t=1}^T f_t(\boldsymbol{\theta}_t)$ and its minimum value, i.e.,

$$R_T = \sum_{t=1}^T f_t(\boldsymbol{\theta}_t) - \min_{\boldsymbol{\theta} \in \mathcal{X}} \sum_{t=1}^T f_t(\boldsymbol{\theta}) = \sum_{t=1}^T \left(f_t(\boldsymbol{\theta}_t) - f_t(\boldsymbol{\theta}^*)\right). \tag{4.1}$$

With further assumption of bounded gradients, we establish convergence analysis of ADAPLUS to ensure the bound of regret as follows.

**Theorem 1.** *Let* $\alpha_t = \alpha/\sqrt{t}$, $\beta_1, \beta_2 \in (0, 1)$, $\gamma = \beta_1/\sqrt{\beta_2} \in (0, 1)$ *and* $\beta_{1t} \leq \beta_1, \forall\, t \in [T]$. *Assume that the convex feasible set* $\mathcal{X}$ *has a bounded diameter* $D_\infty$, *i.e.,* $\forall\, \boldsymbol{x}, \boldsymbol{y} \in \mathcal{X}$, $\|\boldsymbol{x} - \boldsymbol{y}\|_\infty \leq D_\infty$. *Also, assume that loss functions* $f_t(\cdot)$ *are convex and have bounded gradients, i.e.,* $\exists\, G_\infty > 0$, $\forall\, \boldsymbol{\theta} \in \mathcal{X}$ *and* $t \in [T]$, $\|\nabla f_t(\boldsymbol{\theta})\|_\infty \leq G_\infty$. *For* $\boldsymbol{\theta}_t$ *generated using the* ADAPLUS *(Algorithm 2), we have the following regret bound:*

$$\begin{aligned}
R_T \leq &\frac{D_\infty^2}{2\alpha(1-\beta_1)} \sum_{i=1}^d \sqrt{(\hat{v}_{T,i} + \Delta)T} + \frac{\alpha(1+\beta_1)\sqrt{1+\log T}}{(1-\beta_1)^2(1-\gamma)\sqrt{1-\beta_2}} \sum_{i=1}^d \|g_{1:T,i}\|_2 \\
&+ \frac{D_\infty^2}{2(1-\beta_1)} \sum_{t=1}^T \sum_{i=1}^d \frac{\beta_{1t}}{\alpha_t} \sqrt{\hat{v}_{t,i} + \Delta}.
\end{aligned} \tag{4.2}$$

This theorem leads directly to the following two corollaries. We will provide the proof of Theorem 1 and Corollary 1, 2 in Appendix A.

**Corollary 1.** *Under the conditions in Theorem 1, and supposing* $\beta_{1t} = \beta_1/t$, $\beta_{1t} \geq 0$, *we have the following regret bound:*

$$\begin{aligned}
R_T \leq &\frac{D_\infty^2}{2\alpha(1-\beta_1)} \sum_{i=1}^d \sqrt{(\hat{v}_{T,i} + \Delta)T} + \frac{\alpha(1+\beta_1)\sqrt{1+\log T}}{(1-\beta_1)^2(1-\gamma)\sqrt{1-\beta_2}} \sum_{i=1}^d \|g_{1:T,i}\|_2 \\
&+ \frac{\beta_1 d D_\infty^2 \sqrt{(1+\rho)G_\infty}}{2\alpha(1-\beta_1)(1-\lambda)^2}.
\end{aligned} \tag{4.3}$$

**Corollary 2.** *Under the conditions in Corollary 1, we further have the following regret bound:*

$$R_T \leq \frac{d D_\infty^2 \sqrt{(1+\rho)G_\infty T}}{2\alpha(1-\beta_1)} + \frac{\alpha(1+\beta_1)dG_\infty\sqrt{(1+\log T)T}}{(1-\beta_1)^2(1-\gamma)\sqrt{1-\beta_2}} + \frac{\beta_1 d D_\infty^2 \sqrt{(1+\rho)G_\infty}}{2\alpha(1-\beta_1)(1-\lambda)^2}. \tag{4.4}$$

*which means* $R_T = \widetilde{O}(\sqrt{T})$, *where* $\widetilde{O}(\cdot)$ *donates the omission of logarithmic factors.*

# 5 EXPERIMENTS

## 5.1 EXPERIMANTAL SETTINGS

In this section, we conduct full experiments on classical tasks both in CNN and RNN architectures, and compare ADAPLUS with popular algorithms, such as SGD/SGDM, ADAM and AMSGRAD to evaluate performance. Unless otherwise specified, ADAPLUS that we utilized in practice is based on the implementation of AMSGRAD, i.e., $\widehat{\boldsymbol{v}}_t = \max(\widehat{\boldsymbol{v}}_{t-1}, \boldsymbol{v}_t)$ is applied, as is described in Algorithm 2. Overall experiments are summarized in Table 1.

In the field of computer vision, we consider three different architectures on the standard CIFAR-10/100 dataset(Krizhevsky et al., 2009). We use VGGNet-16(Simonyan & Zisserman, 2015),

Table 1: An Overview of Experiments.

| Tasks | Architectures | Datasets | Framework |
|---|---|---|---|
| CV | **VGGNet-16**(Simonyan & Zisserman, 2015)
**ResNet-50**(He et al., 2016)
**DenseNet-121**(Huang et al., 2017) | CIFAR-10/100 | Torch |
| NLP | **OpenNMT**(Klein et al., 2017) | IWSLT15 | Tensorflow |

ResNet-50(He et al., 2016) and DenseNet-121(Huang et al., 2017), where there was a clear distinction in the number of layers to better identify different behaviors of algorithms. We employ the fixed budget of 200 epochs and multiply the learning rates by 0.1 at 100th and 150th epochs. In the Neural Machine Translation experiment(**?**), we chose the learning rate attenuation scheme *luong234*, which means after 2/3 num train steps, we start halving the learning rate for 4 times before finishing. Using the same parameter settings as the benchmark (Luong et al., 2017), we run a total of 12000 steps on Titan XP.

For reproducibility, we provide the codes anonymously in `https://anonfiles.com/daV7Ed6enb/AdaPlus_zip` and put full details of experiments in the Appendix C.

## 5.2 CONVOLUTIONAL NEURAL NETWORK ON CIFAR-10/100

In this section, we train three different models with various numbers of layers to accurately compare the performance of optimizers. We also conducted experiments both on CIFAR-10 and CIFAR-100 to examine the impacts of complex tasks and different datasets on the behavior of optimizers. We report training loss and test accuracy in the following figures.

In general, we notice that in deep neural network architecture the final performance of SGDM is significantly better than ADAM and AMSGRAD, which shows issues in generalization of ADAM-like algorithms once again. Additionally, to our surprise, our approach ADAPLUS outperforms SGDM in the various models and tasks almost at all the time step (with significant differences or a bit similar).

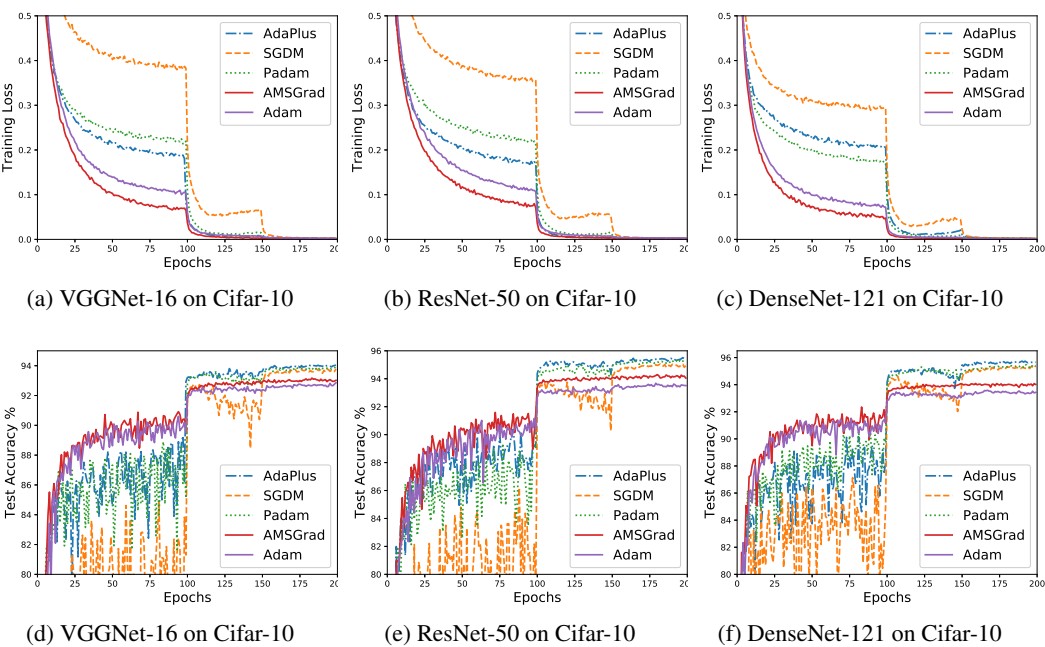

(a) VGGNet-16 on Cifar-10 (b) ResNet-50 on Cifar-10 (c) DenseNet-121 on Cifar-10

(d) VGGNet-16 on Cifar-10 (e) ResNet-50 on Cifar-10 (f) DenseNet-121 on Cifar-10

Figure 2: Training loss and test accuracy of different architectures on Cifar-10.

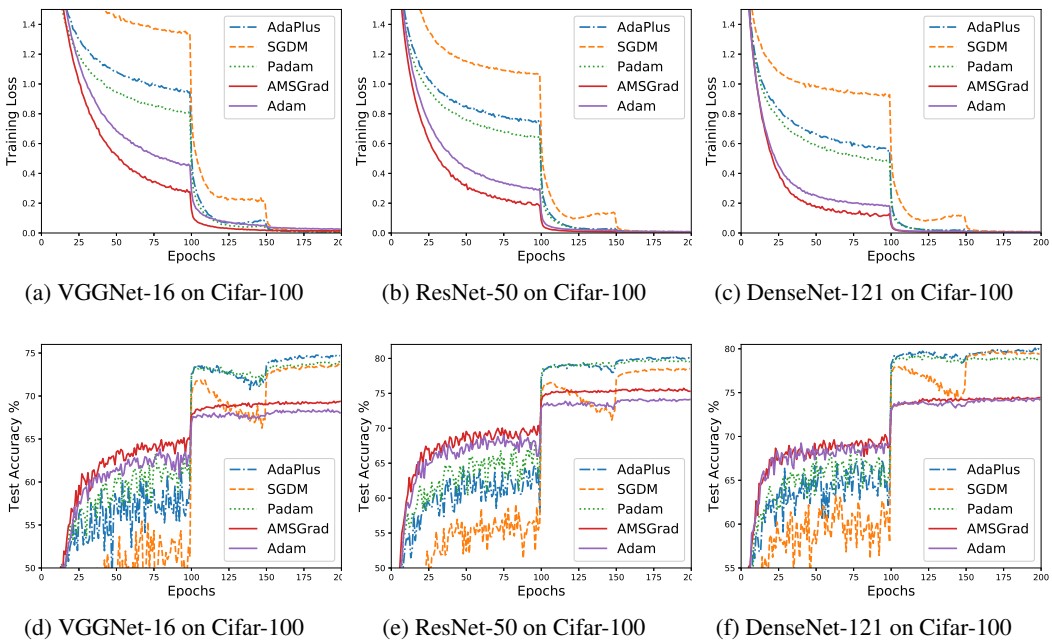

Figure 3: Training loss and test accuracy of different architectures on Cifar-100.

**Summary of behaviors.** Empirically, the behaviors of various algorithms match with what we analyze in Section 2.2. Primarily, ADAPLUS has similar behaviors to PADAM, showing faster convergence and less volatility than SGDM in the early stage. The results of PADAM and SGDM are mostly identical, sometimes SGDM wins while other times PADAM wins. However, ADAPLUS has achieved better performance than both of them due to the introduction of offset term $\Delta$. In addition, both ADAM and AMSGRAD perform poorly in the end, although their early training losses decline rapidly and their convergence is faster.

**Summary of performances.** For VGGNet-16, ResNet-50 and DenseNet-121, ADAPLUS has achieved test accuracies of 0.26%, 0.46%, and 0.15% higher than SGDM in Cifar-10 tasks; while in Cifar-100 tasks, the gaps are about 1.18%, 1.62% and 0.54%. In terms of architectures, the most significant improvement of ADAPLUS incurs in ResNet-50, followed by VGGNet-16 and DenseNet-121. ADAPLUS has achieved an accuracy of about 80% in ResNet-50 on CIFAR-100, which is a significant improvement over SGDM. In terms of complexity of tasks, when the tasks are more complex, ADAPLUS can gain better promotions, since architectures have been improved more by ADAPLUS on CIFAR-100 than on CIFAR-10.

### 5.3 NEURAL MACHINE TRANSLATION

In this section, we evaluate the performance of ADAPLUS in a classic NLP task Neural Machine Translation (NMT)(Klein et al., 2017). Due to the particularity of NLP problems, adaptive optimization algorithms and their variants do not perform as well as SGD; therefore, many researchers prefer to use fine-tuned SGD(Wilson et al., 2017).

OpenNMT has standard benchmarks with open source codes(Luong et al., 2017), from which we select to conduct the IWSLT15 English-to-Vietnam task with the same settings as its benchmark. Since there is no TensorFlow implementation for PADAM, we will not consider it but focus on the comparison of ADAPLUS with the state-of-the-art benchmark. As is generally agreed, standard SGD with $lr = 1.0$ can achieve the best average performance of 26.1 in the model with $beam = 10$. We apply ADAPLUS to the same task in this section. Besides, we experiment in the Vietnam-to-English task of 15k steps in Appendix D, using the same settings as ADASHIFT(Zhou et al., 2019), which also yields better results than SGD.

Figure 4 shows that AMSGRAD's performance is worse than ADAM and that despite the fluctuations of ADAM are smaller, SGD performs indeed better. Therefore, we do not apply AMSGRAD in ADAPLUS, i.e., the sixth line of Algorithm 2 is annotated. Our NMT experiment further proves that ADAPLUS is more suitable for NLP problems with recurrent neural network model and sparse data set. As is shown in Figure 4, ADAPLUS is almost always superior to all other methods. Finally, it has achieved the best BLEU slightly higher than SGD, as shown in Table 2.

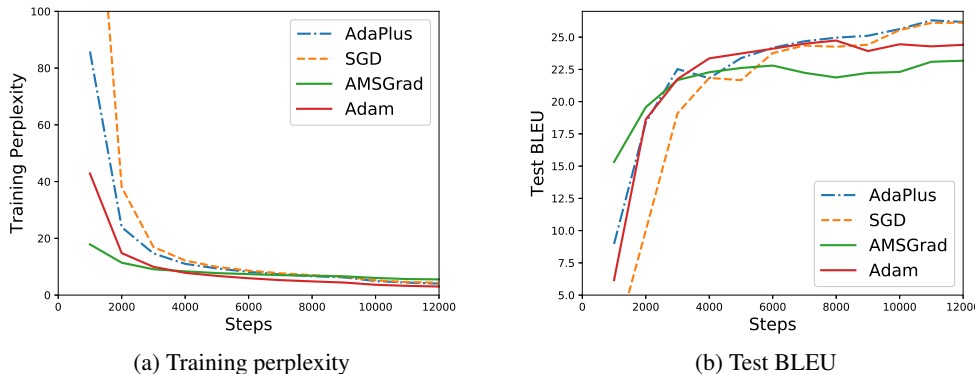

(a) Training perplexity          (b) Test BLEU

Figure 4: Training perplexity and test BLEU on NMT.

Table 2: Best BLEU for 12k steps on IWSLT15 English-to-Vietnam.

| Optimizer | SGD | ADAM | AMSGRAD | ADAPLUS |
|---|---|---|---|---|
| **Best BLEU** | 26.12 | 24.74 | 23.17 | **26.31** |

## 6 FUTURE WORK

Although we present a generic framework and an explicit algorithm with excellent performance, there would be more elegant forms remaining for us to explore in the future.

1. **The choice of function** $\Phi(\cdot)$. For instance, we can replace the square function with $\log(\cdot)$ or $\tanh(\cdot)$ and so on. Also, the square function can be tuned into a $p$-th power function to apply a PADAM-like grid search.

2. **The choice of** $L^p$ **Norm parameter** $p$. Different $L^p$ Norm parameter can be tried to get better numerical performance.

## 7 CONCLUSION

This work proposes a novel generic framework, in which we explicitly analyze different behaviors brought by various types of $\Phi(\cdot)$, such as the constant function in SGDM, the linear function in ADAM, the concave function in PADAM and the concave function with offset term in ADAPLUS. With better adaptivity as is demonstrated, ADAPLUS has achieved remarkable superior results in both CNN and RNN experiments. Our main contributions can be summarized as follows:

- **A generic framework**. Combining ADAM-like algorithms with SGD-like algorithms, the adaptive adjustment function $\Phi(\cdot)$ suggests a generic framework we desired.

- **Explicit analysis of different algorithm behaviors by** $\Phi(\cdot)$. Based on the explicit analysis of $\Phi(\cdot)$, we explain the different behaviors of different algorithms, which is a fundamental reason never mentioned before.

- **The proposal of a concave function with offset term**. In ADAPLUS, we propose an offset term $\Delta$, which can further avoid extreme actual step sizes based on the concave function and achieve superior performance.

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

## A    PROOF OF CONVERGENCE

### A.1    PROOF OF THEOREM 1

*Proof.* Since all $f_t(\cdot)$ are convex, which means $\forall\, \mathbf{x}, \mathbf{y} \in \mathcal{X}$,

$$f_t(\mathbf{y}) \geq f_t(\mathbf{x}) + \nabla f_t(\mathbf{x})^\top (\mathbf{y} - \mathbf{x}). \tag{A.1}$$

Then, we have

$$
\begin{aligned}
R_T &= \sum_{t=1}^{T} \left( f_t\left(\boldsymbol{\theta}_t\right) - f_t\left(\boldsymbol{\theta}^*\right) \right) \\
&\leq \sum_{t=1}^{T} \left\langle \mathbf{g}_t, \boldsymbol{\theta}_t - \boldsymbol{\theta}^* \right\rangle.
\end{aligned} \tag{A.2}
$$

Consider the explicit update formula in Algorithm 2, and use the notation $\widehat{\mathbf{V}}_t = \mathrm{diag}\left(\widehat{\mathbf{v}}_t\right)$. Here, we can get $\boldsymbol{\theta}_{t+1} = \Pi_{\mathcal{X},(\widehat{\mathbf{V}}_t+\Delta)^{\frac{1}{2}}} \left( \boldsymbol{\theta}_t - \alpha_t(\widehat{\mathbf{V}}_t + \Delta)^{-\frac{1}{2}} \cdot \mathbf{m}_t \right)$.

We use $\boldsymbol{Q}_t \triangleq \sqrt{\widehat{\mathbf{V}}_t + \Delta} \in \mathcal{S}_+^d$ to donate the matrix for simplification. Besides, we can find that $\boldsymbol{Q}_t \triangleq \Phi(\widehat{\mathbf{V}}_t)$ is the general formation in Algorithm 1, as long as it satisfies the requirement of being positive definite matrices, the following derivation suits as well.

For the definition of $\boldsymbol{\theta}^*$, it holds that $\Pi_{\mathcal{X},\boldsymbol{Q}}\left(\boldsymbol{\theta}^*\right) = \boldsymbol{\theta}^*$, $\forall \boldsymbol{\theta}^* \in \mathcal{X}$. Using Lemma 1, we have

$$\left\| \boldsymbol{Q}_t^{\frac{1}{2}} \left( \boldsymbol{\theta}_{t+1} - \boldsymbol{\theta}^* \right) \right\|_2^2 \leq \left\| \boldsymbol{Q}_t^{\frac{1}{2}} \left( \boldsymbol{\theta}_t - \alpha_t \boldsymbol{Q}_t^{-1} \cdot \mathbf{m}_t - \boldsymbol{\theta}^* \right) \right\|_2^2. \tag{A.3}$$

Expanding the squared norm on the right side of the inequality and further expanding the momentum term, there is

$$
\begin{aligned}
&\left\| \boldsymbol{Q}_t^{\frac{1}{2}} \left( \boldsymbol{\theta}_{t+1} - \boldsymbol{\theta}^* \right) \right\|_2^2 \\
&\leq \left\| \boldsymbol{Q}_t^{\frac{1}{2}} \left( \boldsymbol{\theta}_t - \boldsymbol{\theta}^* \right) \right\|_2^2 + \alpha_t^2 \left\| \boldsymbol{Q}_t^{-\frac{1}{2}} \mathbf{m}_t \right\|_2^2 - 2\alpha_t \left\langle \mathbf{m}_t, \boldsymbol{\theta}_t - \boldsymbol{\theta}^* \right\rangle \\
&= \left\| \boldsymbol{Q}_t^{\frac{1}{2}} \left( \boldsymbol{\theta}_t - \boldsymbol{\theta}^* \right) \right\|_2^2 + \alpha_t^2 \left\| \boldsymbol{Q}_t^{-\frac{1}{2}} \mathbf{m}_t \right\|_2^2 - 2\alpha_t \left\langle \beta_{1t}\mathbf{m}_{t-1} + (1 - \beta_{1t})\,\mathbf{g}_t, \boldsymbol{\theta}_t - \boldsymbol{\theta}^* \right\rangle
\end{aligned} \tag{A.4}
$$

Rearranging the terms in the above inequality, the following first inequality holds,

$$
\begin{aligned}
&\left\langle \mathbf{g}_t, \boldsymbol{\theta}_t - \boldsymbol{\theta}^* \right\rangle \\
&\leq \frac{1}{2\alpha_t\left(1-\beta_{1t}\right)} \left[ \left\| \boldsymbol{Q}_t^{\frac{1}{2}}\left(\boldsymbol{\theta}_t - \boldsymbol{\theta}^*\right) \right\|_2^2 - \left\| \boldsymbol{Q}_t^{\frac{1}{2}}\left(\boldsymbol{\theta}_{t+1} - \boldsymbol{\theta}^*\right) \right\|_2^2 \right] + \frac{\alpha_t}{2\left(1-\beta_{1t}\right)} \cdot \left\| \boldsymbol{Q}_t^{-\frac{1}{2}}\mathbf{m}_t \right\|_2^2 \\
&\quad - \frac{\beta_{1t}}{1-\beta_{1t}} \left\langle \mathbf{m}_{t-1}, \boldsymbol{\theta}_t - \boldsymbol{\theta}^* \right\rangle \\
&\leq \frac{1}{2\alpha_t\left(1-\beta_{1t}\right)} \left[ \left\| \boldsymbol{Q}_t^{\frac{1}{2}}\left(\boldsymbol{\theta}_t - \boldsymbol{\theta}^*\right) \right\|_2^2 - \left\| \boldsymbol{Q}_t^{\frac{1}{2}}\left(\boldsymbol{\theta}_{t+1} - \boldsymbol{\theta}^*\right) \right\|_2^2 \right] + \frac{\alpha_t}{2\left(1-\beta_{1t}\right)} \cdot \left\| \boldsymbol{Q}_t^{-\frac{1}{2}}\mathbf{m}_t \right\|_2^2 \\
&\quad + \frac{\beta_{1t}\alpha_t}{2\left(1-\beta_{1t}\right)} \cdot \left\| \boldsymbol{Q}_{t-1}^{-\frac{1}{2}}\mathbf{m}_{t-1} \right\|_2^2 + \frac{\beta_{1t}}{2\alpha_t\left(1-\beta_{1t}\right)} \left\| \boldsymbol{Q}_{t-1}^{\frac{1}{2}}\left(\boldsymbol{\theta}_t - \boldsymbol{\theta}^*\right) \right\|_2^2.
\end{aligned} \tag{A.5}
$$

and the last inequality holds due to Cauchy-Schwarz and Young's inequality.

Combining (A.2) with (A.5), we have

$$\sum_{t=1}^{T} \left[ f_t \left( \boldsymbol{\theta}_t \right) - f_t \left( \boldsymbol{\theta}_t^* \right) \right]$$

$$\leq \sum_{t=1}^{T} \left\{ \frac{1}{2\alpha_t \left( 1 - \beta_{1t} \right)} \left[ \left\| \boldsymbol{Q}_t^{\frac{1}{2}} \left( \boldsymbol{\theta}_t - \boldsymbol{\theta}^* \right) \right\|_2^2 - \left\| \boldsymbol{Q}_t^{\frac{1}{2}} \left( \boldsymbol{\theta}_{t+1} - \boldsymbol{\theta}^* \right) \right\|_2^2 \right] + \frac{\alpha_t}{2 \left( 1 - \beta_{1t} \right)} \cdot \left\| \boldsymbol{Q}_t^{-\frac{1}{2}} \mathbf{m}_t \right\|_2^2 \right.$$

$$\left. + \frac{\beta_{1t} \alpha_t}{2 \left( 1 - \beta_{1t} \right)} \cdot \left\| \boldsymbol{Q}_{t-1}^{-\frac{1}{2}} \mathbf{m}_{t-1} \right\|_2^2 + \frac{\beta_{1t}}{2\alpha_t \left( 1 - \beta_{1t} \right)} \left\| \boldsymbol{Q}_{t-1}^{\frac{1}{2}} \left( \boldsymbol{\theta}_t - \boldsymbol{\theta}^* \right) \right\|_2^2 \right\}.$$

$$\tag{A.6}$$

As is separately proven in Lemma 2, Lemma 3 and Lemma 4, we can further get the desired regret bound.

$$R_T = \sum_{t=1}^{T} \left[ f_t \left( \boldsymbol{\theta}_t \right) - f_t \left( \boldsymbol{\theta}_t^* \right) \right]$$

$$\leq \frac{D_\infty^2}{2\alpha \left( 1 - \beta_1 \right)} \sum_{i=1}^{d} \sqrt{\left( \hat{v}_{T,i} + \Delta \right) T} + \frac{\alpha \left( 1 + \beta_1 \right) \sqrt{1 + \log T}}{\left( 1 - \beta_1 \right)^2 \left( 1 - \gamma \right) \sqrt{1 - \beta_2}} \sum_{i=1}^{d} \left\| g_{1:T,i} \right\|_2 \tag{A.7}$$

$$+ \frac{D_\infty^2}{2 \left( 1 - \beta_1 \right)} \sum_{t=1}^{T} \sum_{i=1}^{d} \frac{\beta_{1t}}{\alpha_t} \sqrt{\hat{v}_{t,i} + \Delta}$$

$\square$

## A.2 PROOF OF COROLLARY 1

*Proof.*
Since we assume that $\beta_{1t} = \beta_1 \lambda^t$, $\lambda \in (0, 1)$, we have

$$\frac{D_\infty^2}{2 \left( 1 - \beta_1 \right)} \sum_{t=1}^{T} \sum_{i=1}^{d} \frac{\beta_{1t}}{\alpha_t} \sqrt{\hat{v}_{t,i} + \Delta}$$

$$= \frac{D_\infty^2}{2 \left( 1 - \beta_1 \right)} \sum_{t=1}^{T} \sum_{i=1}^{d} \frac{\beta_1 \lambda^{t-1}}{\alpha} \sqrt{\left( \hat{v}_{t,i} + \Delta \right) t} \tag{A.8}$$

Besides, due to the definition of $\hat{v}_{t,i}$, we have

$$\hat{v}_{t,i} + \Delta = \left( 1 - \beta_2 \right) \sum_{j=1}^{t} \beta_2^{t-j} |g_{j,i}| + \Delta \leq \left( 1 - \beta_2 \right) \left( \sum_{j=1}^{t} \beta_2^{t-j} |g_{j,i}| + \frac{\Delta}{1 - \beta_2} \right). \tag{A.9}$$

Since $\Delta > 0$ is a given constant, which is chosen to be relatively small enough, it always holds that $\exists \rho > 0$, s.t., $\Delta \leq \rho \cdot G_\infty$. Then, we can further improve the result in (A.9), $\forall t \in [T]$,

$$\hat{v}_{t,i} + \Delta \leq \left( 1 - \beta_2 \right) \left( \sum_{j=1}^{t} \beta_2^{t-j} \right) G_\infty + \rho \cdot G_\infty \leq \left( 1 + \rho \right) G_\infty, \tag{A.10}$$

which leads to

$$\frac{D_\infty^2}{2 \left( 1 - \beta_1 \right)} \sum_{t=1}^{T} \sum_{i=1}^{d} \frac{\beta_{1t}}{\alpha_t} \sqrt{\hat{v}_{t,i} + \Delta}$$

$$\leq \frac{\beta_1 D_\infty^2 \sqrt{\left( 1 + \rho \right) G_\infty}}{2\alpha \left( 1 - \beta_1 \right)} \sum_{t=1}^{T} \sum_{i=1}^{d} \lambda^{t-1} \sqrt{t} \tag{A.11}$$

$$\leq \frac{\beta_1 d D_\infty^2 \sqrt{\left( 1 + \rho \right) G_\infty}}{2\alpha \left( 1 - \beta_1 \right) \left( 1 - \lambda \right)^2}.$$

Submitting (A.11) into (A.7), we extend the conclusion in Theorem 1 to Corollary 1.

$$
\begin{aligned}
R_T &= \sum_{t=1}^{T} \left[ f_t \left( \boldsymbol{\theta}_t \right) - f_t \left( \boldsymbol{\theta}_t^* \right) \right] \\
&\leq \frac{D_\infty^2}{2\alpha \left( 1 - \beta_1 \right)} \sum_{i=1}^{d} \sqrt{\left( \hat{v}_{T,i} + \Delta \right) T} + \frac{\alpha \left( 1 + \beta_1 \right) \sqrt{1 + \log T}}{\left( 1 - \beta_1 \right)^2 \left( 1 - \gamma \right) \sqrt{1 - \beta_2}} \sum_{i=1}^{d} \| g_{1:T,i} \|_2 \\
&\quad + \frac{\beta_1 d D_\infty^2 \sqrt{\left( 1 + \rho \right) G_\infty}}{2\alpha \left( 1 - \beta_1 \right) \left( 1 - \lambda \right)^2}
\end{aligned}
\tag{A.12}
$$

$\square$

## A.3    PROOF OF COROLLARY 2

*Proof.*
Primarily, as is proved in (A.10), it holds that

$$
\hat{v}_{T,i} + \Delta \leq \left( 1 - \beta_2 \right) \left( \sum_{j=1}^{T} \beta_2^{T-j} \right) G_\infty + \rho \cdot G_\infty \leq \left( 1 + \rho \right) G_\infty.
\tag{A.13}
$$

Thus, we have

$$
\begin{aligned}
&\frac{D_\infty^2 \sqrt{T}}{2\alpha \left( 1 - \beta_1 \right)} \sum_{i=1}^{d} \sqrt{\hat{v}_{T,i} + \Delta} \\
&\leq \frac{D_\infty^2 \sqrt{T}}{2\alpha \left( 1 - \beta_1 \right)} \sum_{i=1}^{d} \sqrt{\left( 1 + \rho \right) G_\infty} \\
&= \frac{d D_\infty^2 \sqrt{\left( 1 + \rho \right) G_\infty \cdot T}}{2\alpha \left( 1 - \beta_1 \right)}
\end{aligned}
\tag{A.14}
$$

Second, due to the property of convex functions and bounded gradients, we further have

$$
\sum_{i=1}^{d} \| g_{1:T,i} \|_2 \leq \sum_{i=1}^{d} \sqrt{\sum_{t=1}^{T} g_{t,i}^2} \leq d G_\infty \sqrt{T},
\tag{A.15}
$$

Pluging (A.14) and (A.15) back into (A.12), we finally get the result of Corollary 2

$$
\begin{aligned}
R_T &= \sum_{t=1}^{T} \left[ f_t \left( \boldsymbol{\theta}_t \right) - f_t \left( \boldsymbol{\theta}_t^* \right) \right] \\
&\leq \frac{d D_\infty^2 \sqrt{\left( 1 + \rho \right) G_\infty T}}{2\alpha \left( 1 - \beta_1 \right)} + \frac{\alpha \left( 1 + \beta_1 \right) d G_\infty \sqrt{\left( 1 + \log T \right) T}}{\left( 1 - \beta_1 \right)^2 \left( 1 - \gamma \right) \sqrt{1 - \beta_2}} \\
&\quad + \frac{\beta_1 d D_\infty^2 \sqrt{\left( 1 + \rho \right) G_\infty}}{2\alpha \left( 1 - \beta_1 \right) \left( 1 - \lambda \right)^2},
\end{aligned}
\tag{A.16}
$$

which means $R_T = \widetilde{O}(\sqrt{T})$, where $\widetilde{O}(\cdot)$ donates the omission of logarithmic factors.

$\square$

## B    AUXILIARY LEMMAS

### B.1    PROOF OF LEMMA 1

**Lemma 1** (McMahan & Streeter (2010); Reddi et al. (2018); Chen & Gu (2018)). *For any $\boldsymbol{Q} \in \mathcal{S}_+^d$ and convex feasible set $\mathcal{X} \subset \mathbb{R}^d$, suppose $\boldsymbol{u_1} = \arg\min_{\boldsymbol{x} \in \mathcal{X}} \| \boldsymbol{Q}^{1/2} (\boldsymbol{x} - \boldsymbol{z_1}) \|_2$ and $\boldsymbol{u_2} = \arg\min_{\boldsymbol{x} \in \mathcal{X}} \| \boldsymbol{Q}^{1/2} (\boldsymbol{x} - \boldsymbol{z_2}) \|_2$ then we have $\| \boldsymbol{Q}^{1/2} (\boldsymbol{u_1} - \boldsymbol{u_2}) \|_2 \leq \| \boldsymbol{Q}^{1/2} (\boldsymbol{z_1} - \boldsymbol{z_2}) \|_2$.*

We will not provide the proof here, since it will be exactly the same as the reference and has been proved many times.

## B.2 PROOF OF LEMMA 2

**Lemma 2.** *Under the conditions in Theorem 1, we have*

$$\sum_{t=1}^{T} \left\{ \frac{\alpha_t}{2\left(1-\beta_{1t}\right)} \left[ \left\|\boldsymbol{Q}_t^{-\frac{1}{2}}\mathbf{m}_t\right\|^2 + \beta_{1t}\left\|\boldsymbol{Q}_{t-1}^{-\frac{1}{2}}\mathbf{m}_{t-1}\right\|^2 \right] \right\} \leq \frac{\alpha\left(1+\beta_1\right)\sqrt{1+\log T}}{\left(1-\beta_1\right)^2\left(1-\gamma\right)\sqrt{1-\beta_2}} \sum_{i=1}^{d} \|g_{1:T,i}\|_2 \,.$$

*Proof.* Similar to (Reddi et al., 2018; Chen & Gu, 2018), we first describe the upper bound of $\sum_{t=1}^{T} \alpha_t \left\|\boldsymbol{Q}_t^{-\frac{1}{2}}\mathbf{m}_t\right\|^2 = \sum_{t=1}^{T} \sum_{i=1}^{d} \frac{\alpha_t \cdot m_{t,i}^2}{\sqrt{\widehat{v}_{t,i}+\Delta}}$ in the following derivation, since we use $\boldsymbol{Q}_t \triangleq \sqrt{\widehat{\mathbf{V}}_t + \Delta} \in \mathcal{S}_+^d$ to donate the update rule.

$$
\begin{aligned}
&\sum_{t=1}^{T} \alpha_t \left\|\boldsymbol{Q}_t^{-\frac{1}{2}}\mathbf{m}_t\right\|^2 \\
&= \sum_{t=1}^{T} \sum_{i=1}^{d} \frac{\alpha_t \cdot m_{t,i}^2}{\sqrt{\widehat{v}_{t,i}+\Delta}} \\
&= \sum_{t=1}^{T-1} \sum_{i=1}^{d} \frac{\alpha_t \cdot m_{t,i}^2}{\sqrt{\widehat{v}_{t,i}+\Delta}} + \sum_{i=1}^{d} \frac{\alpha_T \cdot m_{T,i}^2}{\sqrt{\widehat{v}_{T,i}+\Delta}} \\
&\leq \sum_{t=1}^{T-1} \sum_{i=1}^{d} \frac{\alpha_t \cdot m_{t,i}^2}{\sqrt{\widehat{v}_{t,i}+\Delta}} + \sum_{i=1}^{d} \frac{\alpha_T \cdot m_{T,i}^2}{\sqrt{v_{T,i}+\Delta}} \\
&= \sum_{t=1}^{T-1} \sum_{i=1}^{d} \frac{\alpha_t \cdot m_{t,i}^2}{\sqrt{\widehat{v}_{t,i}+\Delta}} + \frac{\alpha}{\sqrt{T}} \sum_{i=1}^{d} \frac{\left(\sum_{j=1}^{T}\left(1-\beta_{1j}\right)\beta_1^{T-j}g_{j,i}\right)^2}{\sqrt{\left(1-\beta_2\right)\sum_{j=1}^{T}\beta_2^{T-j}|g_{j,i}|+\Delta}} \\
&\leq \sum_{t=1}^{T-1} \sum_{i=1}^{d} \frac{\alpha_t \cdot m_{t,i}^2}{\sqrt{\widehat{v}_{t,i}+\Delta}} + \frac{\alpha}{\sqrt{T}\left(1-\beta_2\right)} \sum_{i=1}^{d} \frac{\left(\sum_{j=1}^{T}\beta_1^{T-j}|g_{j,i}|^{\frac{1}{2}}\right)\left(\sum_{j=1}^{T}\beta_1^{T-j}|g_{j,i}|^{\frac{3}{2}}\right)}{\sqrt{\sum_{j=1}^{T}\beta_2^{T-j}|g_{j,i}|+\frac{\Delta}{1-\beta_2}}}
\end{aligned}
\tag{B.1}
$$

These several equalities holds due to the update rule and definitions of $\alpha_t$, $v_{t,i}$ and $m_{t,i}$. Also, the first inequality holds due to the definition of $\widehat{v}_{t,i}$, while the second inequality holds due to $\beta_{1t} \leq \beta_1, \forall\, t \in [T]$ and Cauchy-Schwarz inequality. Since $\sum_{j=1}^{T} \beta_1^{T-j} \leq \frac{1}{1-\beta_1}$ and $f_t(\cdot)$ has bounded gradients, we have

$$
\begin{aligned}
&\sum_{t=1}^{T} \alpha_t \left\|\boldsymbol{Q}_t^{-\frac{1}{2}}\mathbf{m}_t\right\|^2 \\
&\leq \sum_{t=1}^{T-1} \sum_{i=1}^{d} \frac{\alpha_t \cdot m_{t,i}^2}{\sqrt{\widehat{v}_{t,i}+\Delta}} + \frac{\alpha\sqrt{G_\infty}}{\left(1-\beta_1\right)\sqrt{T\left(1-\beta_2\right)}} \sum_{i=1}^{d} \frac{\sum_{j=1}^{T}\beta_1^{T-j}|g_{j,i}|^{\frac{3}{2}}}{\sqrt{\sum_{j=1}^{T}\beta_2^{T-j}|g_{j,i}|+\frac{\Delta}{1-\beta_2}}} \\
&\leq \sum_{t=1}^{T-1} \sum_{i=1}^{d} \frac{\alpha_t \cdot m_{t,i}^2}{\sqrt{\widehat{v}_{t,i}+\Delta}} + \frac{\alpha\sqrt{G_\infty}}{\left(1-\beta_1\right)\sqrt{T\left(1-\beta_2\right)}} \sum_{i=1}^{d} \frac{\sum_{j=1}^{T}\beta_1^{T-j}|g_{j,i}|^{\frac{3}{2}}}{\sqrt{\sum_{j=1}^{T}\beta_2^{T-j}|g_{j,i}|}} \\
&\leq \sum_{t=1}^{T-1} \sum_{i=1}^{d} \frac{\alpha_t \cdot m_{t,i}^2}{\sqrt{\widehat{v}_{t,i}+\Delta}} + \frac{\alpha\sqrt{G_\infty}}{\left(1-\beta_1\right)\sqrt{T\left(1-\beta_2\right)}} \sum_{j=1}^{T} \sum_{i=1}^{d} \frac{\beta_1^{T-j}|g_{j,i}|^{\frac{3}{2}}}{\sqrt{\beta_2^{T-j}|g_{j,i}|}} \\
&\leq \sum_{t=1}^{T-1} \sum_{i=1}^{d} \frac{\alpha_t \cdot m_{t,i}^2}{\sqrt{\widehat{v}_{t,i}+\Delta}} + \frac{\alpha\sqrt{G_\infty}}{\left(1-\beta_1\right)\sqrt{T\left(1-\beta_2\right)}} \sum_{j=1}^{T} \sum_{i=1}^{d} \gamma^{T-j}|g_{j,i}| \,,
\end{aligned}
\tag{B.2}
$$

where the second inequality is indeed strictly less for $\Delta > 0$, the third inequality holds due to the property of concave function. At last, as is assumed, $\gamma = \frac{\beta_1}{\sqrt{\beta_2}} \in (0,1)$ is introduced.

We further expand $\sum\limits_{t=1}^{T-1} \alpha_t \left\| \boldsymbol{Q}_t^{-\frac{1}{2}} \mathbf{m}_t \right\|^2 = \sum\limits_{t=1}^{T-1} \sum\limits_{i=1}^{d} \frac{\alpha_t \cdot m_{t,i}^2}{\sqrt{\hat{v}_{t,i}+\Delta}}$ in the same way, which leads to

$$
\begin{aligned}
&\sum_{t=1}^{T} \alpha_t \left\| \boldsymbol{Q}_t^{-\frac{1}{2}} \mathbf{m}_t \right\|^2 \\
&\leq \frac{\alpha\sqrt{G_\infty}}{(1-\beta_1)\sqrt{1-\beta_2}} \sum_{t=1}^{T} \frac{1}{\sqrt{t}} \sum_{j=1}^{t} \sum_{i=1}^{d} \gamma^{t-j} |g_{j,i}| \\
&= \frac{\alpha\sqrt{G_\infty}}{(1-\beta_1)\sqrt{1-\beta_2}} \sum_{i=1}^{d} \sum_{t=1}^{T} \frac{|g_{t,i}|}{\sqrt{t}} \sum_{j=t}^{T} \gamma^{j-t} \\
&\leq \frac{\alpha\sqrt{G_\infty}}{(1-\beta_1)\sqrt{1-\beta_2}} \sum_{i=1}^{d} \sum_{t=1}^{T} \frac{|g_{t,i}|}{(1-\gamma)\sqrt{t}} \\
&= \frac{\alpha\sqrt{G_\infty}}{(1-\gamma)(1-\beta_1)\sqrt{1-\beta_2}} \sum_{i=1}^{d} \sum_{t=1}^{T} \frac{|g_{t,i}|}{\sqrt{t}} \\
&\leq \frac{\alpha\sqrt{G_\infty}}{(1-\gamma)(1-\beta_1)\sqrt{1-\beta_2}} \sum_{i=1}^{d} \|g_{1:T,i}\|_2 \sqrt{\sum_{t=1}^{T} \frac{1}{t}} \\
&\leq \frac{\alpha\sqrt{G_\infty(1+\log T)}}{(1-\gamma)(1-\beta_1)\sqrt{1-\beta_2}} \sum_{i=1}^{d} \|g_{1:T,i}\|_2
\end{aligned}
\tag{B.3}
$$

Similarly, with $\sum\limits_{j=t}^{T} \gamma^{j-t} \leq \frac{1}{1-\gamma}$, the second inequality holds. The third inequality also holds by Cauchy-Schwarz inequality, while the last inequality holds due to the bound on harmonic sum: $\sum\limits_{t=1}^{T} \frac{1}{t} \leq 1 + \log T$.

Thus, we finally get

$$
\begin{aligned}
&\sum_{t=1}^{T} \left\{ \frac{\alpha_t}{2(1-\beta_{1t})} \left[ \left\| \boldsymbol{Q}_t^{-\frac{1}{2}} \mathbf{m}_t \right\|^2 + \beta_{1t} \left\| \boldsymbol{Q}_{t-1}^{-\frac{1}{2}} \mathbf{m}_{t-1} \right\|^2 \right] \right\} \\
&\leq \frac{\alpha(1+\beta_1)\sqrt{G_\infty(1+\log T)}}{2(1-\gamma)(1-\beta_1)^2\sqrt{1-\beta_2}} \sum_{i=1}^{d} \|g_{1:T,i}\|_2 .
\end{aligned}
\tag{B.4}
$$

$\square$

### B.3  Proof of Lemma 3

**Lemma 3.** *Under the conditions in Theorem 1, we have*

$$
\sum_{t=1}^{T} \frac{1}{2\alpha_t(1-\beta_{1t})} \left[ \left\| \boldsymbol{Q}_t^{\frac{1}{2}}(\boldsymbol{\theta}_t - \boldsymbol{\theta}^*) \right\|_2^2 - \left\| \boldsymbol{Q}_t^{\frac{1}{2}}(\boldsymbol{\theta}_{t+1} - \boldsymbol{\theta}^*) \right\|_2^2 \right] \leq \frac{D_\infty^2}{2\alpha(1-\beta_1)} \sum_{i=1}^{d} \sqrt{(\hat{v}_{T,i}+\Delta)T}.
$$

*Proof.* Consider

$$
\sum_{t=1}^{T} \frac{1}{2\alpha_t (1 - \beta_{1t})} \left[ \left\| \boldsymbol{Q}_t^{\frac{1}{2}} (\boldsymbol{\theta}_t - \boldsymbol{\theta}^*) \right\|_2^2 - \left\| \boldsymbol{Q}_t^{\frac{1}{2}} (\boldsymbol{\theta}_{t+1} - \boldsymbol{\theta}^*) \right\|_2^2 \right]
$$

$$
\leq \sum_{t=1}^{T} \sum_{i=1}^{d} \frac{\sqrt{\hat{v}_{t,i} + \Delta}}{2\alpha_t (1 - \beta_1)} \left[ (\theta_{t+1,i} - \theta_i^*)^2 - (\theta_{t,i} - \theta_i^*)^2 \right]
$$

$$
= \frac{1}{2(1 - \beta_1)} \left[ \sum_{i=1}^{d} \frac{\sqrt{\hat{v}_{1,i} + \Delta} (\theta_{1,i} - \theta_i^*)^2}{\alpha_1} + \sum_{l=2}^{T} \sum_{i=1}^{d} \left( \frac{\sqrt{\hat{v}_{t,i} + \Delta}}{\alpha_t} - \frac{\sqrt{\hat{v}_{t-1,i} + \Delta}}{\alpha_{t-1}} \right) (\theta_{t,i} - \theta_i^*)^2 \right]
$$

$$
\leq \frac{D_\infty^2}{2(1 - \beta_1)} \left[ \sum_{i=1}^{d} \frac{\sqrt{\hat{v}_{1,i} + \Delta}}{\alpha_1} + \sum_{t=2}^{T} \sum_{i=1}^{d} \left( \frac{\sqrt{\hat{v}_{t,i} + \Delta}}{\alpha_t} - \frac{\sqrt{\hat{v}_{t-1,i} + \Delta}}{\alpha_{t-1}} \right) \right]
$$

$$
= \frac{D_\infty^2}{2(1 - \beta_1)} \sum_{i=1}^{d} \frac{\sqrt{\hat{v}_{T,i} + \Delta}}{\alpha_T}
$$

$$
= \frac{D_\infty^2}{2\alpha(1 - \beta_1)} \sum_{i=1}^{d} \sqrt{(\hat{v}_{T,i} + \Delta) T},
$$

(B.5)

where the first inequality is based on $\beta_{1t} \leq \beta_1$, and the second inequality is due to the definition of a bounded feasible set.

$\square$

## B.4 PROOF OF LEMMA 4

**Lemma 4.** *Under the conditions in Theorem 1, we have*

$$
\sum_{t=1}^{T} \frac{\beta_{1t}}{2\alpha_t (1 - \beta_{1t})} \left\| \boldsymbol{Q}_{t-1}^{\frac{1}{2}} (\boldsymbol{\theta}_t - \boldsymbol{\theta}^*) \right\|_2^2 \leq \frac{D_\infty^2}{2(1 - \beta_1)} \sum_{t=1}^{T} \sum_{i=1}^{d} \frac{\beta_{1t}}{\alpha_t} \sqrt{\hat{v}_{t,i} + \Delta}.
$$

*Proof.* Consider

$$
\sum_{t=1}^{T} \frac{\beta_{1t}}{2\alpha_t (1 - \beta_{1t})} \left\| \boldsymbol{Q}_{t-1}^{\frac{1}{2}} (\boldsymbol{\theta}_t - \boldsymbol{\theta}^*) \right\|_2^2
$$

$$
\leq \sum_{t=1}^{T} \sum_{i=1}^{d} \frac{\beta_{1t} \cdot \sqrt{\hat{v}_{t,i} + \Delta}}{2\alpha_t (1 - \beta_1)} (\theta_{t,i} - \theta_i^*)^2
$$

(B.6)

$$
\leq \frac{D_\infty^2}{2(1 - \beta_1)} \sum_{t=1}^{T} \sum_{i=1}^{d} \frac{\beta_{1t}}{\alpha_t} \sqrt{\hat{v}_{t,i} + \Delta},
$$

where the first inequality is based on $\beta_{1t} \leq \beta_1$ and the non-decreasing property of $\hat{v}_{t,i}$, and the second inequality is due to the definition of a bounded feasible set. $\square$

## C EXPERIMENTAL SETTINGS

### C.1 CIFAR CLASSIFICATION

- $\beta_1$ **and** $\beta_2$**.** We conduct experiments in $(\beta_1, \beta_2) = \{(0.9, 0.99), (0.9, 0.999)\}$. For ADAM and AMSGRAD, we use $(0.9, 0.99)$; while for SGDM, PADAM and ADAPLUS, we use $(0.9, 0.999)$.

- **Learning Rate.** For ADAM and AMSGRAD, we do grid search in $\{0.001, 0.0001\}$. For SGDM, PADAM and ADAPLUS, we do grid search in $\{0.05, 0.1, 0.2, 0.3\}$, and most optimal results are using $lr = 0.1$.

- **Learning Rate Schedule.** We conduct a learning rate schedule of decaying the learning rate by 0.1 at the 100th and 150th epoches in all 200 epoches.

- **Offset.** For ADAPLUS, we do grid search in $\Delta = \{1/8, 1/4, 1/2\}$ to get best results.

- **Weight Decaying.** For ADAM and AMSGRAD, we set $weight\_decaying = 1e-4$; while for SGDM, PADAM and ADAPLUS, we set $weight\_decaying = 5e-4$.

- **Best Setting.** For ADAPLUS, we fine-tune the hyper-parameters and achieve best results when $lr = 0.1, \Delta = 1/8$ in VGGNet-Cifar100, ResNet-Cifar100 and DenseNet-Cifar10, $lr = 0.2, \Delta = 1/2$ in VGGNet-Cifar10, $lr = 0.1, \Delta = 1/4$ in ResNet-Cifar10 and $lr = 0.3, \Delta = 1/2$ in DenseNet-Cifar10.

## C.2 NEURAL MACHINE TRANSLATION

- **$\beta_1$ and $\beta_2$.** We conduct experiments in $(\beta_1, \beta_2) = \{(0.9, 0.99), (0.9, 0.999)\}$. For ADAM and AMSGRAD, we use $(0.9, 0.99)$; while for SGDM, PADAM and ADAPLUS, we use $(0.9, 0.999)$.

- **Learning Rate.** For ADAM and AMSGRAD, we do grid search in $\{0.001, 0.0001\}$. we searched learning rates in $\{0.1, 0.5, 1.0, 2.0\}$ for SGD, and get the optimal result when $lr = 1.0$.

- **Learning Rate Schedule.** We conduct a learning rate scheme *luong234*, which means after 2/3 num train steps, we start halving the learning rate for 4 times in all 12000 steps.

- **Offset.** For ADAPLUS, we do grid search in $\Delta = \{1/2, 1, 2\}$ to get best results.

- **Other Details.** We keep all other details the same with the benchmark(Luong et al., 2017), there is also a json file in our codes to demonstrate parameter settings.

- **Best Setting.** For ADAPLUS, we achieve the best result when $lr = 2.0, \Delta = 2.0$ or $lr = 1.0, \Delta = 1.0$.

## C.3 ADDITIONAL EXPERIMENTS

- **$\beta_1$ and $\beta_2$.** We conduct experiments in $(\beta_1, \beta_2) = \{(0.9, 0.99), (0.9, 0.999)\}$. For ADAM and AMSGRAD, we use $(0.9, 0.99)$; while for SGDM, PADAM and ADAPLUS, we use $(0.9, 0.999)$.

- **Learning Rate.** For ADAM and AMSGRAD, we do grid search in $\{0.001, 0.0001\}$. we searched learning rates in $\{0.1, 0.5, 1.0, 2.0\}$ for SGD, and get the optimal result when $lr = 1.0$.

- **Learning Rate Schedule.** We conduct a learning rate scheme *self*, which means after 8000 train steps, we start halving the learning rate every 1000 steps within 15000 steps of training.

- **Offset.** For ADAPLUS, we do grid search in $\Delta = \{1/8, 1/4, 1/2\}$ to get best results.

- **Other Details.** We keep all other details the same with ADASHIFT(Zhou et al., 2019).

- **Best Setting.** For ADAPLUS, we achieve the best result when $lr = 0.5, \Delta = 0.125$.

# D ADDITIONAL EXPERIMENTS IN NMT

We also conduct another NMT experiment on IWSLT15. Unlike Section 5.3, we apply a different learning rate schedule, extend the total number of training steps to 15000 steps, and change the task into Vietnam-to-English. We conduct a learning rate scheme *self*, which means after 8000 train steps, we start halving the learning rate every 1000 steps within the total 15000 steps of training. We have fine-tuned different optimizers and compare their best performance, where the results of ADAM and AMSGRAD are similar to that in (Zhou et al., 2019). The experimental results still show that ADAPLUS has significant advantages over other algorithms, both in terms of convergence speed and final performance.

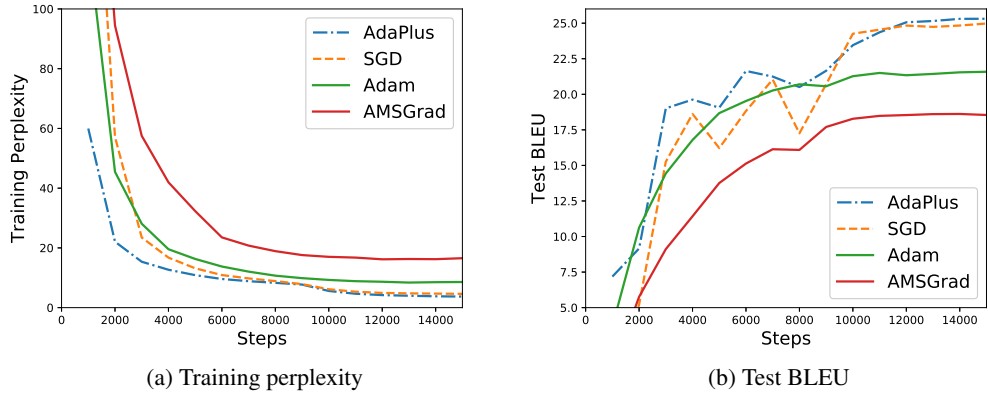

(a) Training perplexity

(b) Test BLEU

Figure 5: Training perplexity and test BLEU on NMT.

Table 3: Best BLEU for 15k steps on IWSLT15 Vietnam-to-English.

| Optimizer | SGD | ADAM | AMSGRAD | ADAPLUS |
|---|---|---|---|---|
| **Best BLEU** | 25.02 | 21.68 | 18.92 | **25.35** |

