# OpenReview forum: "ADA+: A GENERIC FRAMEWORK WITH MORE ADAPTIVE EXPLICIT ADJUSTMENT FOR LEARNING RATE"
_ICLR.cc/2020/Conference — Reject_

### Official Review · AnonReviewer1 · 2019-10-23
**Official Blind Review #1**

**Rating:** 3

**Review:**

This work proposes a modification to the ADAM optimizer by introducing an adjustment function, which consists of a square root function and an extra parameter delta. Although the modification is very simple and easy to implement, the theoretical analysis is weak and the empirical performances of the proposed method are similar to the previous adaptive methods.

Here are my main concerns of the current paper:
1. The presentation is a little bit confusing in the motivation section. According to equation (2.1), the $L^p$ norm of $\mathbf{g}_t$ is defined as a vector. However, it seems that the authors treat $\|\mathbf{g}_t\|_p$ as a scalar when presenting the intuition of the proposed method in Figure 1.

2. I do not understand why Figure 1 says the proposed method is better than Padam. It seems to me that if Padam choose a specific p, it can recover the proposed $\Phi$ function, or even better than the proposed method. Therefore, I do not think the intuition of the proposed method is correct.

3. For the convergence analysis, the authors only consider the convex setting, which I think is meaningless. Because the proposed method is designed for training neural networks, such convergence guarantee in convex setting is not enough. There exist some work such as [1] have proved the convergence guarantee of the adaptive algorithms including Padam in the nonconvex setting.

4. There is one missing baseline Yogi [2] in the current paper.

5. For experimental results, the performance of the proposed method is very similar to the Padam. Due to the close formulation of the proposed method and Padam, it seems to me that the proposed method is  just a more careful hyperparameter tuning process.

6. In NMT experiments, why there is no Padam baseline? In addition, the authors should also report the test perplexity to validate the generalization performance of the proposed optimizer.

7. To fully evaluate the performance of the proposed method, the authors should at least conduct an experiment on the task of language model.

Minor comments:
There is an unknown citation in section 5.1.

Reference:
[1]. Zhou, Dongruo, et al. "On the convergence of adaptive gradient methods for nonconvex optimization." arXiv preprint arXiv:1808.05671 (2018).
[2]. Zaheer, Manzil, et al. "Adaptive methods for nonconvex optimization." Advances in Neural Information Processing Systems. 2018.

**Experience Assessment:**

I have read many papers in this area.

**Review Assessment: Checking Correctness Of Derivations And Theory:**

I assessed the sensibility of the derivations and theory.

**Review Assessment: Checking Correctness Of Experiments:**

I carefully checked the experiments.

**Review Assessment: Thoroughness In Paper Reading:**

I read the paper thoroughly.

---

### Official Review · AnonReviewer3 · 2019-10-23
**Official Blind Review #3**

**Rating:** 3

**Review:**

This work proposes a general framework for adaptive algorithms, and presents a specific form: ADAPLUS. In the theory part, this work gives convergence analysis of ADAPLUS. For experiments, this work analyzes several algorithm's empirical performances including SGDM, ADAM, AMSGRAD, PADAM, ADAPLUS on CV and NLP tasks.

1. This paper's analysis is not solid enough to support its claim. Since this paper gives a general framework and claims that offset term can achieve superior performance, it is better to give the convergence analysis of general algorithm in the framework and discuss the benefit of the offset term theoretically. Actually, the author gives almost the same theoretical result as ADAM type algorithms, from which I did not see the advantage of using ADAPLUS.

2. And the experiment shows that ADAPLUS performs on par with PADAM on CV task.
I wonder why the authors did not give the experimental result of PADAM on NLP task?

3.The notation in this paper is quite confusing. In page 2 definition (2.1), v is a vector so the L^p norm of g is actually a vector. But in Figure on page 2, the paper treat it as a scalar and I cannot understand it without further explanation from the authors.

Therefore, I think this work doesn't make enough contribution and the novelty is not enough for ICLR standard.


**Experience Assessment:**

I have read many papers in this area.

**Review Assessment: Checking Correctness Of Derivations And Theory:**

I assessed the sensibility of the derivations and theory.

**Review Assessment: Checking Correctness Of Experiments:**

I assessed the sensibility of the experiments.

**Review Assessment: Thoroughness In Paper Reading:**

I read the paper thoroughly.

---

### Official Review · AnonReviewer4 · 2019-10-27
**Official Blind Review #4**

**Rating:** 1

**Review:**

This paper proposes a new variation on adaptive learning rate algorithm that builds on a prior work Padam, which is also a concurrent submission to this conference. The method is empirically validated through through various domains and neural networks architecture. Though the empirical results are extensive, I am leaning towards reject because (1) The method is a very small variation; no insight is provided for why. (2) The theory is not very useful in justifying the method. (3) The empirical results are weak.

(1) This work builds on the prior work Padam, which is not a well justified algorithm. Same criticisms hence can be made to this paper. I don't see any strong arguments from the paper that well justifies the methodology either.
The problems are listed below:

Here's what I can find of the authors' interpretation of the prior work Padam:

"The internal cause is that a concave function is applied rather than the linear function in ADAM.
Once ε is extremely small and |g| ∈ (0, ε), the mapping value of Φ(·) would be much
larger in PADAM than in ADAM; therefore, PADAM can adapt to larger learning rate α,
thus flexibly adapting to the variable learning rate scheme."

(i) I don't think phrasing the cause as a result of concavity gives arise to any new insights. Simply, what this is trying to say is that Φ(·) is large when |g| is small, that's why PADAM can adapt to large learning rate.

Building on Padam, the paper further justifies the proposed method by:

" This form of Φ(·) not only directly inherits advantages of PADAM,
as is depicted in Figure 1, but also makes a better guarantee for larger learning rates. The
offset ∆ makes sure that Φ(·) can altogether avoid the extreme situation. Even when
|g| → 0, a more extensive learning rate αt is allowed. "

(ii) The paper inherits a serious problem from Padam, that is to assume large learning rate is important for learning rate decay. Padam didn't explain this, nor does this paper. So the justification is very weak.
(iii) The introduction of the offset is not novel. Just as the author noted, this is almost the same term epsilon in original updates. It's not rare that one also tunes epsilon for some optimization problems.

(2) Continuing the last point, the theoretical analysis focus on the convergence proof of the algorithm. The proof is not new. Theory also is not useful in justifying the method.

(3) The method seems not to be able to beat the previous baseline Padam, which makes it questionable as a practical algorithm.

**Experience Assessment:**

I have published in this field for several years.

**Review Assessment: Checking Correctness Of Derivations And Theory:**

I carefully checked the derivations and theory.

**Review Assessment: Checking Correctness Of Experiments:**

I assessed the sensibility of the experiments.

**Review Assessment: Thoroughness In Paper Reading:**

I read the paper thoroughly.

---

### Decision · Program_Chairs · 2019-12-19

**Decision:**

Reject

**Comment:**

In this paper, the authors proposed a general framework, which uses an explicit function as an adjustment to the actual learning rate, and presented a more adaptive specific form Ada+. Based on this framework, they analyzed various behaviors brought by different types of the function. Empirical experiments on benchmarks demonstrate better performance than some baseline algorithms. The main concern of this paper is: (1) lack of justification or interpretation for the proposed framework; (2) the performance of the proposed algorithm is on a par with Padam; (3) missing comparison with some other baselines on more benchmark datasets. Plus, the authors did not submit response.  I agree with the reviewers’ evaluation.